# Salt Solubilization Coupled with Membrane Filtration-Impact on the Structure/Function of Chickpea Compared to Pea Protein

**DOI:** 10.3390/foods12081694

**Published:** 2023-04-19

**Authors:** Brigitta P. Yaputri, Fan Bu, Baraem P. Ismail

**Affiliations:** Food Science and Nutrition Department, University of Minnesota, 1334 Eckles Ave, Saint Paul, MN 55108, USA

**Keywords:** pea protein isolate, chickpea protein isolate, salt extraction coupled with ultrafiltration, scaled-up production, structural characteristics, functional properties

## Abstract

The demand for pulse proteins as alternatives to soy protein has been steeply increasing over the past decade. However, the relatively inferior functionality compared to soy protein is hindering the expanded use of pulse proteins, namely pea and chickpea protein, in various applications. Harsh extraction and processing conditions adversely impact the functional performance of pea and chickpea protein. Therefore, a mild protein extraction method involving salt extraction coupled with ultrafiltration (SE-UF) was evaluated for the production of chickpea protein isolate (ChPI). The produced ChPI was compared to pea protein isolate (PPI) produced following the same extraction method in terms of functionality and feasibility of scaling. Scaled-up (SU) ChPI and PPI were produced under industrially relevant settings and evaluated in comparison to commercial pea, soy, and chickpea protein ingredients. Controlled scaled-up production of the isolates resulted in mild changes in protein structural characteristics and comparable or improved functional properties. Partial denaturation, modest polymerization, and increased surface hydrophobicity were observed in SU ChPI and PPI compared to the benchtop counterparts. The unique structural characteristics of SU ChPI, including its ratio of surface hydrophobicity and charge, contributed to superior solubility at both a neutral and acidic pH compared to both commercial soy protein and pea protein isolates (cSPI and cPPI) and significantly outperformed cPPI in terms of gel strength. These findings demonstrated both the promising scalability of SE-UF and the potential of ChPI as a functional plant protein ingredient.

## 1. Introduction

The demand for plant protein ingredients has considerably increased in recent years due to their low production cost, positive environmental impact, nutritional value, and health benefits. Accordingly, the plant protein ingredient market is expected to reach $3 billion by 2031 [1]. Yellow field pea (*Pisum sativum* L.) and chickpea (*Cicer arietinum* L.) protein ingredients are major contenders in the plant protein market, with an expected market share of $555 million by 2029 and $158 million by 2032, respectively [2,3]. Both pea and chickpea protein ingredients have been used as soy protein alternatives in high-protein food and beverages, including plant-based meat products [4,5].

Although pea and chickpea proteins have a similar profile to soy protein, both have relatively inferior functional properties, specifically gelation, emulsification, and solubility [6,7,8,9]. The functionality limitations hinder the expanded use of pea and chickpea proteins in various applications. The commercially available pea protein isolate generally has relatively poor functional properties due to excessive protein denaturation and polymerization attributed to harsh extraction and processing conditions [7,8,10]. Recently, it was shown that mild and controlled extraction and processing conditions can preserve pea protein structure and functionality [7].

While both the pea protein isolate (PPI) and the pea protein concentrate (PPC) are widely available commercially, chickpea protein in the market is mostly available in the form of chickpea protein concentrate (ChPC). Chickpea protein isolate (ChPI), on the other hand, is a rare commercial commodity. Both PPC and ChPC are produced by air classification that does not involve wet or thermal processing, contrary to the production of isolates. The most common commercial process for the production of plant protein isolates is alkaline extraction to separate the protein from the starch and fiber, followed by isoelectric point precipitation (AE-IEP) to isolate and purify the protein [11]. However, high alkalinity results in a high degree of protein denaturation and aggregation, which negatively impact functionality [7,8,12,13,14]. Mild extraction conditions, including salt solubilization coupled with membrane filtration, have been shown to preserve the protein structure and functionality [7]. While pea protein ingredient production has been studied extensively, limited research has been reported on the impact of extraction conditions on chickpea protein structural and functional properties at both bench-scale and under industrially relevant settings [9,15,16,17,18].

In our previous study, we determined that salt solubilization coupled with membrane filtration is an industrially feasible approach to producing a functional PPI [7]. Additionally, protein structural characteristics were preserved and functional properties were better than those of a commercial PPI produced following AE-IEP. To the best of our knowledge, there is no research on ChPI production following salt extraction coupled with ultrafiltration (SE-UF) at bench nor at a pilot scale. Most studies optimized ChPI production following AE-IEP [9,16,17,18], or alkaline extraction (at pH 9) coupled with ultrafiltration [18]. In another study, salt extraction coupled with dialysis was used to produce ChPI, which exhibited good functionality, yet was still inferior to soy protein [9]. The limited functionality could be attributed to the residual high-salt content that shielded the charge on the surface of the protein [19,20]. Ultrafiltration coupled with diafiltration or dialysis would be a more efficient strategy to remove excess residual salt. Therefore, SE-UF has the potential to produce ChPI with preserved structural characteristics and good functionality compared to currently available chickpea protein ingredients.

To evaluate the scalability and transferability of SE-UF for ChPI production, the objectives of this study were to (1) evaluate SE-UF conditions for the production of ChPI with acceptable purity and yield, (2) evaluate the scalability of the SE-UF process, and (3) determine the structural and functional properties of PPI and ChPI compared to commercial sources. 

## 2. Materials and Methods

### 2.1. Materials 

Yellow field pea flour (20% protein) and commercial pea protein concentrate (cPPC) (52.4% protein, 5.14% ash), FYPP-55, were provided by AGT Foods (Regina, SK, Canada). Defatted chickpea flour (26.8% protein), Artesa™ Chickpea Flour 20 M, and commercial chickpea protein concentrate (cChPC) (56.4% protein, 5.27% ash), Artesa™ Chickpea Protein, were provided by Nutriati (Henrico, VA, USA). Commercial soy protein isolate (cSPI) (90.1% protein, 4.16% ash), ProFam^®^ 974, and commercial pea protein isolate (cPPI) (79.5% protein, 5.61% ash), ProFam^®^ 580, were kindly provided by Archer Daniels Midland (ADM) (Decatur, IL, USA). Samples were stored at −20 °C when not in use. Vivaflow^®^ ultrafiltration membrane crossflow cassettes (3 kDa cut-off) were purchased from Sartorius™ (Gottingen, Germany). SnakeSkin™ dialysis tubing (3.5 kDa cut off) and Sudan Red 7B were purchased from Thermo Fisher Scientific™ (Waltham, MA, USA). Precision Plus molecular weight marker, Criterion™ TGX™ 4–20% precast gels, Laemmli 4X loading buffer, 10X Tris/Glycine/SDS running buffer, and Imperial™ Protein Stain were purchased from Bio-Rad Laboratories, Inc. (Hercules, CA, USA). 8-anilino-1-napthalenesulfonic acid ammonium salt (ANS), and 2-mercaptoethanol (BME) were purchased from Sigma-Aldrich (St. Louis, MO, USA). For size-exclusion high-performance liquid chromatography (SE-HPLC), SuperdexTM 200 Increase 10/300 GL Prepacked TricornTM Column, and gel filtration LMW and HMW calibration kits were purchased from Cytiva (Marlborough, MA, USA). All other analytical-grade reagents and lab supplies were purchased from Sigma-Aldrich or Thermo Fisher Scientific.

### 2.2. Selection of Salt Solubilization Conditions for the Production of ChPI

In our previous study, the salt extraction conditions (0.5 M NaCl, 1 h of solubilization at room temperature) were selected for bench and pilot scale production of PPI based on protein yield and purity [7]. These salt extraction conditions were, therefore, used as the baseline for selecting the extraction conditions for ChPI production. Several studies reported that an elevated temperature could enhance protein solubilization, thereby contributing to a relatively higher protein yield [21,22]. Two temperatures (23 °C and 50 °C) and three salt concentrations (0.5 M, 0.75 M, and 1 M) were tested to determine the combination that might result in enhanced chickpea protein solubilization. In triplicate, chickpea flour was solubilized in 0.5, 0.75, or 1.0 M NaCl solution prepared with double deionized water (DDW) (1:20 *w*/*v*) and stirred for 1 h at its initial pH (~6.8) under room temperature (23 °C) or at 50 °C. The solution was then centrifuged at 5000× *g* for 10 min to separate the insoluble components and the supernatant was collected. The protein content of the supernatant was determined following the Dumas method (AOAC 990.03) using a LECO^®^ FP828 nitrogen analyzer (LECO, St. Joseph, MI, USA), with a protein conversion factor of 6.25. Under all the tested conditions, up to ~65% of the protein in the starting flour was retrieved in the collected supernatant, with a slightly lower percent when 1.0 M NaCl was used. Therefore, the lowest salt concentration (0.5 M NaCl) coupled with solubilization at room temperature was selected for ChPI production.

### 2.3. Benchtop Production of PPI and ChPI

Benchtop production of PPI and ChPI was performed following the SE-UF method outlined by Hansen et al. [7] and the SE-UF conditions selected based on protein yield, respectively. Pea or chickpea flour was dispersed in a 0.5 M NaCl solution three times (1:20 *w*/*v*) and was stirred for 1 h at its initial pH (~6.8) and at room temperature (23 °C). The solution was then centrifuged at 5000× *g* for 30 min to precipitate insoluble components. The residual pellet was then lyophilized and its protein content was later determined for mass balance calculations. The supernatant, containing the solubilized protein, was collected, and the pH was adjusted to 7.0. Prior to ultrafiltration, a vacuum filtration step was included to filter the protein solution and remove small insoluble particles that could clog the ultrafiltration membrane. The solution was then subjected to crossflow ultrafiltration (UF) using the Sartorius Vivaflow^®^ 200 system, followed by dialysis as described by Hansen et al. [7] to further remove residual salt and small molecular weight sugars to enhance protein purity. Components with a molecular weight larger than the membrane pore size (3 kDa) were retained and recirculated to the feed reservoir. Components smaller than 3 kDa passed through the membrane and were collected as permeate in the waste container. The protein solution was concentrated down to 50 mL. After concentration, the solution was diafiltered with 50 mL of DDW six times (300 mL total) to further purify the protein. At the end of diafiltration, the solution was concentrated down to 25 mL. To flush the remaining protein solution left on the membrane and increase protein yield, approximately 25 mL of DDW was pumped into the system. After filtration and dialysis, the samples were lyophilized. The protein yield and purity of PPI and ChPI were determined by the Dumas method. Ash content (AOAC method 942.05), moisture content (AOAC method 926.08), and fat content (AOAC method 922.06) were also determined.

### 2.4. Pilot Plant Scale-Up Production of PPI and ChPI

Pilot scale SE-UF was performed in the Joseph J. Warthensen Food Processing Center, University of Minnesota, to produce scaled-up (SU) isolates, SU PPI, and SU ChPI, following the process reported by Hansen et al. [7], with some modifications. To produce SU PPI and SU ChPI, pea or chickpea flour was dispersed in water with 0.5 M NaCl (1:20 *w*/*v*) in a jacketed tank. The solution was stirred for 1 h at room temperature at its initial pH (~6.8). To separate the protein supernatant from the starch slurry, the solution was passed through a horizontal decanter centrifuge (Westfalia Separator AG, 1 gal/min, GEA Westfalia Separator Group Gmbh, Oelde, Germany) coupled with a desludging disc centrifuge (Westfalia SB7, 1 gal/min, GEA Westfalia Separator Group Gmbh, Oelde, Germany). The starch slurry was then re-solubilized in water with 0.5 M NaCl (1:5 *w*/*v*), stirred for 30 min, and passed through a second round of decanter and clarifier to optimize protein extraction. The protein supernatants from the first and second solubilization were combined and the pH was adjusted to 7.0 followed by ultrafiltration (3.5 kDa cut-off) to remove residual salt. To monitor salt removal and total loss, the total solids (TS) (%) of the permeate was regularly checked using a CEM AVC-80 Microwave Moisture/Solids Balance Analyzer (CEM, Charlotte, NC, USA). When the TS of the permeate reached 0.0%, the protein retentate solution was concentrated until its TS reached 8–10%, pasteurized, homogenized, and spray dried using an SPX Flow Anhydro Spray Dryer (9.5% TS, 180 °C inlet, 90 °C outlet, 9 L/h) with a wheel type atomizer (24,500 rpm) (SPX Flow Inc., Charlotte, NC, USA). The residual protein left on the membrane was flushed out and collected separately. Since the flushed protein only contained approximately 5% TS, an evaporation step was performed to concentrate the solution to 8% TS prior to pasteurization, homogenization, and spray drying. The non-evaporated protein retentate was referred to as “high solids” (HS), while the portion that underwent evaporation was referred to as “low solids” (LS). The spray-dried HS and LS were combined after structural characterization screening showed no significant differences between the two fractions. The protein, ash, moisture, and fat content of SU PPI and SU ChPI were determined as described above. SU isolates were stored at −20 °C when not in use.

### 2.5. Color Measurement

The color (L * a * b *) of benchtop, SU, and commercial protein samples was measured three times as described by Bu et al. [8] using a Chroma Meter CR-221 (Minolta Camera Co., Osaka, Japan). The total color difference (ΔE) between each produced isolate and its respective commercial ingredient was also calculated.

### 2.6. Protein Structural Characterization

#### 2.6.1. Protein Profiling by Gel Electrophoresis

The protein profile of all samples was monitored using sodium dodecyl sulfate polyacrylamide gel electrophoresis (SDS-PAGE), as described by Laemmli [23] and modified by Boyle et al. [21]. Precision Plus™ MW standard and protein samples (5 μL, containing approximately 50 μg protein) in Laemmli buffer with and without a reducing agent (βME) were loaded onto Criterion™TGX™ 4–20% precast Tris-HCl gradient gel and electrophorized at 200 V. The gels were then stained with Imperial Protein Stain™, de-stained with DDW, and scanned using the Molecular Imager Gel Doc XR system (Bio-Rad Laboratories).

#### 2.6.2. Protein Denaturation State

The denaturation temperature and enthalpy of all samples were determined, in triplicate, using DSC (Mettler Toledo, Columbus, OH, USA), following the method outlined by Bu et al. [8]. The thermograms were manually integrated using Mettler Toledo’s STARe Software version 11.00.

#### 2.6.3. Surface Properties of Protein Ingredients

To determine the surface charge, zeta potential (ζ) was measured, in triplicate, using a dynamic light scattering instrument (Malvern Panalytical, Malvern, UK) as described by Bu et al. [8], with no modifications. The surface hydrophobicity of all samples was determined using the spectrofluorometric method outlined by Boyle et al. [21] and modified by Bu et al. [8].

### 2.7. Protein Functional Characterization

#### 2.7.1. Protein Solubility

The protein solubility of all protein samples was determined using the method outlined by Boyle et al. [21] and modified by Bu et al. [8]. Protein solutions (5% protein *w*/*v*) were prepared, in triplicate, at pH 7 and at pH 3.4 to assess the suitability for high protein neutral as well as acidic beverages. The protein solubility was measured at both room temperature and post-thermal treatment (80 °C for 30 min) using the Dumas method.

#### 2.7.2. Gel Strength and Water Holding Capacity (WHC)

Thermally-induced gels were prepared as outlined by Bu et al. [8], with modifications in the protein concentration and heating time. In triplicate, 10 mL protein solutions (15% and 20% protein, *w*/*v*) were stirred for 2 h and adjusted to pH 7. The 15% protein solutions were heated for 10 min in a water bath at 95 °C (±2 °C), whereas the 20% protein solutions were heated for 20 min. After cooling, gel strength was measured using a TA-XT Plus Texture Analyzer (Stable Micro Systems LTD, Surrey, UK) following the same parameters outlined by Bu et al. [8]. The force (N) required to rupture the gel was recorded as gel strength. WHC of all samples (15% protein concentration, *w*/*v*) was measured as described by Boyle et al. [21], without modification.

#### 2.7.3. Emulsification Properties

Emulsion capacity (EC, at 1% protein in DDW, *w*/*v*), activity index (EAI), and stability (ES) of all samples were determined at pH 7, in triplicate, according to the methods outlined by Boyle et al. [21] and Bu et al. [8].

### 2.8. Statistical Analysis 

Analysis of variance (ANOVA) was determined using IBM^®^ SPSS^®^ Statistics software version 26 for Windows (International Business Machines Corp., Armonk, NY, USA). Tukey–Kramer multiple means comparison test was used to identify significant differences (*p* ≤ 0.05) among the means of at least three samples. A student’s unpaired t-test was used to test for significant differences (*p* ≤ 0.05) between the means of the two samples.

## 3. Results and Discussion

### 3.1. Impact of Salt Extraction on the Protein Purity and Yield of Benchtop and Scaled-Up ChPI in Comparison to PPI

ChPI had a high protein purity (>90%) similar to that of PPI (Table 1). While Mondor et al. [18] utilized benchtop ultrafiltration to purify chickpea protein, the produced ChPI had lower protein purity (~84% on average), which could be attributed to limited protein solubility under the alkaline extraction conditions used in the absence of salt. In addition, the high MWCO membrane (50 kDa) used in their study likely led to a significant loss of protein. In another study by Karaca et al. [9], PPI and ChPI produced using salt solubilization coupled with dialysis on a bench scale had a lower protein purity (~81%) compared to the isolates produced in this study following SE-UF. Such a difference in protein purity could be attributed not only to the use of dialysis instead of ultrafiltration, but also to the use of a different salt type (K_2_SO_4_) at a low concentration (~0.3 M) for the solubilization of chickpea protein. Salt concentration and related ionic strength have a unique impact on protein solubilization based on the specific protein structure and surface charge. The chosen salt concentration should provide enough ions to stabilize the protein in the aqueous solution (salting in) [24].

While neither Mondor et al. [18] nor Karaca et al. [9] reported the protein yield, Espinosa-Ramírez et al. [25] reported chickpea protein extraction yields of up to 67%. The reported higher protein extraction yield compared to the obtained yield in this study (52%) is attributed to the high alkalinity of the solubilization solution (pH 9.5) used by Espinosa-Ramírez et al. [25], which was detrimental to protein functionality. While the chickpea protein yield obtained in this study is acceptable and comparable to what has been reported for pulse proteins [26], it is significantly (*p* < 0.05) lower than that obtained for PPI (64%), despite following similar extraction conditions. A significantly (*p* < 0.05) higher % residual protein (~34% of the original protein in the flour) was left in the pellet discarded during ChPI production compared to that (18% of the original protein in the flour) discarded during PPI production. This observation could be attributed to the content and structure of the starch [27] and fiber [28] in chickpeas compared to peas, which could have hindered protein solubilization and extraction efficiency. 

The SU production of PPI and ChPI achieved similar protein purity to the benchtop counterparts (Table 1), with minor statistical differences. ChPI had a significantly higher ash content than SU ChPI, which could explain the slightly higher protein purity of SU ChPI. However, SU production of PPI and ChPI resulted in significantly (*p* < 0.05) lower protein yield, 59% and 41%, respectively, mostly due to losses during the decanting step. Nevertheless, SE-UF was successfully scaled up, for the first time, to produce ChPI with high protein purity and relatively low ash content, demonstrating the feasibility of production at an industrial scale. To improve the yield during scaled-up production, enhancement of the decanting step should be targeted in future trials. 

### 3.2. Impact of Extraction Scale on the Color of PPI and ChPI Compared to Commercial Protein Ingredients

The SU PPI and SU ChPI were significantly lighter and more neutral in color compared to their benchtop counterparts (Table 1). Different drying methods (spray drying vs. freeze drying) used to produce SU and benchtop isolates could be mainly responsible for the color differences. The size and morphology of the particles could change the intensity and angle of the reflected light, resulting in different perceptions of color. In general, spray drying produces a more refined powder with a smaller particle size compared to freeze drying [29,30]. Spray-dried particles were reported to have a rounded morphology with the wrinkled surface, which enables the particles to reflect/scatter more light compared to freeze-dried particles that have a smooth surface, and plate-shaped morphology [30,31].

Commercial protein concentrates (cPPC, cChPC), on the other hand, exhibited a significantly lighter color than all the protein isolates, mostly attributed to the higher starch content in the concentrates. When comparing PPI and SU PPI to cPPI, ∆E was modest. Therefore, it can be concluded that the SE-UF used in this study resulted in protein ingredients (SU PPI and SU ChPI) of a similar color profile to commercial counterparts, with slightly lighter and more neutral color compared to cPPI. This observation can be attributed to potentially less browning during the SE-UF process utilized in this study compared to the AE-IEP process used to produce cPPI.

### 3.3. Protein Profile of the Benchtop and Scaled-Up Isolates in Comparison to Commercial Samples

The protein profile of benchtop and scaled-up PPIs and ChPIs was compared to commercial samples under nonreducing and reducing conditions (Figure 1a,b). Under nonreducing conditions (Figure 1a, lanes 4–5), PPI and SU PPI had protein bands corresponding to lipoxygenase (~92 kDa), convicilin (~72 kDa), legumin (~60 kDa), vicilin (13–19, 30–35, and 50 kDa), and albumin (~10 kDa), similar to the pea protein profile reported in other studies [11,15,32,33,34,35]. Under reducing conditions, the disulfide linkages stabilizing the legumin monomers were cleaved, resulting in protein bands corresponding to the acidic and basic legumin subunits at ~40 kDa and ~20 kDa, respectively (Figure 1b, lanes 4 & 5). Similarly, under both nonreducing and reducing conditions (Figure 1, lanes 7 & 8), ChPI and SU ChPI had protein bands corresponding to the major protein components observed in PPI and SU PPI, in agreement with previous reports [25,36]. However, the bands corresponding to legumin monomers (under nonreducing conditions) and legumin acidic and basic subunits (under reducing conditions) were more intense than their counterparts in PPI and SU PPI, with a couple of additional variants that have slightly different molecular weights. A similar protein band pattern of legumin in chickpeas was also observed by Chang et al. [36], Wang et al. [37], Vioque et al. [38], and Papalamprou et al. [39].

The protein profile of PPI and SU PPI, and that of ChPI and SU ChPI, were similar to cPPI and cPPC, and to cChPC, respectively. However, under nonreducing conditions, cPPI showed intense smearing in the upper region of its lane with no apparent legumin band at 60 kDa, indicating a high extent of legumin-involved polymerization (Figure 1a, lane 2). In contrast, there was no such smearing in cPPC (Figure 1a, lane 3). The air classification used to produce cPPC is a mild process [40,41] compared to the wet milling extraction process followed to produce cPPI. The use of a harsh alkaline extraction process induced protein denaturation and subsequent polymerization in cPPI, as was previously discussed by Hansen et al. [7]. Even under reducing conditions, dark bands and residual smearing persisted in the upper region of the cPPI lane (Figure 1a, lane 2), indicating the presence of high molecular weight (HMW) protein polymers that are stabilized by other types of covalent linkages, other than disulfide bonds. Irreversible covalent linkages induced by the Maillard reaction are commonly formed under excessive heat treatment and elevated pH [42,43,44]. These observations confirmed that the conditions used to produce PPI and ChPI at the bench as well as pilot scale were relatively mild, preventing the formation of large polymers that may negatively impact functionality.

However, there was mild smearing observed in the upper region of the SU PPI and SU ChPI lanes compared to those of PPI and ChPI (Figure 1a, lanes 5 & 8 vs. lanes 4 & 7), indicating the presence of some HMW legumin-involved polymers. Under the reducing condition, the smearing was resolved, indicating that the observed protein polymerization was mainly attributed to disulfide linkages (Figure 1b, lanes 5 & 8). The presence of such polymers was thermally induced during evaporation, pasteurization, and spray-drying steps of the scaled-up production. However, based on protein profiling (Figure 2), the formation of these polymers was mostly attributed to the evaporation step applied to the low solids (LS) fractions of SU PPI and SU ChPI. Dark smearing was noted in the upper region of the lanes of the LS fractions compared to those of the high solids (HS) fractions (Figure 2, lanes 3 & 5 compared to lanes 2 & 4), which was mostly resolved under reducing conditions (Figure 2, lanes 7 & 9). Spray-dried LS fractions were mixed with spray-dried HS fractions to produce the final SU isolate, thus explaining the presence of HMW polymers in both SU isolates.

In contrast to cPPC, cChPC had mild smearing in the upper region of its lane (Figure 1a, lane 6), similar to that noted for SU ChPI. Although cChPC was produced via air classification, the initial flour was defatted prior to air classification. The defatting process, while important to reduce the fat content of chickpea flour (7% to less than 1% fat), involves thermal desolventization, which will induce protein denaturation and subsequent polymerization. Having seemingly similar polymerization patterns, both cChPC and SU ChPI potentially may have similar protein functionality. 

### 3.4. Protein Denaturation State of PPI and ChPI as Impacted by Extraction Scale and in Comparison to Commercial Protein Ingredients 

The impact of the extraction scale (benchtop vs. scale-up) on the protein denaturation state of PPI and ChPI was evaluated in comparison to commercial protein ingredients (Table 2). Two distinct denaturation temperatures, corresponding to vicilin and legumin, were observed for benchtop and scaled-up PPI and ChPI, in agreement with previous studies [6,15]. Since the endothermic peaks of vicilin and legumin overlapped, as was also observed by others [39,45], both peaks were integrated as one peak and the total enthalpy of denaturation was obtained (Table 2).

No apparent endothermic peak was observed for commercial protein isolates (cPPI and cSPI), indicating complete protein denaturation due to extensive wet processing conditions. In contrast, legumin and vicilin endothermic peaks were present in both cPPC and cChPC, which underwent air classification. As discussed, air classification is a milder process compared to wet milling [40,41]. However, the enthalpy of denaturation of the concentrates was significantly lower than that of the benchtop as well as the scaled-up PPI and ChPI samples. This observation could be attributed to matrix differences between isolates and concentrates, regardless of the extraction/fractionation conditions [46]. 

The presence of prominent endothermic peaks with a relatively high enthalpy of denaturation in the produced isolates compared to cPPI confirmed that the SE-UF process preserved the protein structure, limiting denaturation (Table 2) and consequent polymerization (Figure 1). Furthermore, ChPI produced on benchtop following AE-IEP [6,47] at high alkalinity had a considerably lower enthalpy (2.5–5.5 J/g) than that of both benchtop and scaled-up ChPI produced by SE-UF in this study. Similarly, benchtop and scaled-up PPI produced following SE-UF had higher denaturation enthalpy than their AE-IEP counterparts [7]. 

When comparing PPI to ChPI, both ChPI and SU ChPI had significantly higher denaturation enthalpy than PPI and SU PPI, respectively (Table 2). Chickpea protein had higher denaturation enthalpy than pea protein, regardless of the extraction scale, most likely due to its relatively higher legumin to vicilin ratio, as noted by SDS-PAGE (Figure 1). On the other hand, benchtop isolates had significantly higher denaturation enthalpy compared to their scaled-up counterparts (Table 2). This observation complimented the protein profiling observations (Figure 1), confirming that the thermal treatments (evaporation, pasteurization, and spray drying) during the scale-up production led to partial protein denaturation and consequent formation of HMW polymers. Partial denaturation may also impact the surface properties of the protein.

### 3.5. Protein Surface Properties of PPI and ChPI as Impacted by Extraction Scale and in Comparison to Commercial Protein Ingredients

Scaled-up isolates had significantly higher surface hydrophobicity than their benchtop counterparts (Table 2), attributed to the partial denaturation incurred during scaling-up production in the pilot plant, as discussed. Upon denaturation, surface hydrophobicity is expected to increase due to protein unfolding and exposure of buried hydrophobic residues [48]. Enhanced surface hydrophobicity drives protein molecules into closer proximity, facilitating different forms of bonding, including disulfide linkages, as noted by SDS-PAGE (Figure 1, lanes 5 & 8 vs. 4 & 7). 

Differences in surface hydrophobicity among the samples were also evaluated in comparison to commercial ingredients. While both SU PPI and cPPI had similar surface hydrophobicity, the latter was completely denatured (Table 2) and excessively polymerized (Figure 1a, lane 2). Maximum surface hydrophobicity is theoretically reached upon complete protein denaturation. However, the excessive polymerization of legumin in cPPI, as observed from protein profiling, likely reduced its measurable surface hydrophobicity. Polymerization of denatured proteins driven by hydrophobic interactions would bury the exposed hydrophobic groups, thus reducing measurable surface hydrophobicity [49]. Thus, due to differences in the extent of polymerization, SU PPI is expected to have better functionality than cPPI, despite having similar surface hydrophobicity. On the other hand, cPPC and benchtop PPI had similar surface hydrophobicity, and both were lacking in HMW polymers (Figure 1a, lanes 3 & 4). Meanwhile, cChPC had significantly higher surface hydrophobicity than both ChPI and SU ChPI. As discussed, this commercial sample was subjected to defatting prior to air classification, causing denaturation and thus exposure of the hydrophobic core. 

In comparing the two different isolates, the pea protein isolate had significantly higher surface hydrophobicity than the chickpea protein isolate, regardless of the extraction scale. This observation could be attributed mostly to inherent differences among the species. The abundance of globulins compared to that of albumins, the ratio of 7S vicilin to 11S legumin, as well as the presence of different subunit variants could all contribute to differences in surface hydrophobicity [50,51]. Karaca et al. [9] similarly reported that PPI had higher surface hydrophobicity than ChPI. In soybeans, it is documented that 11S glycinin has higher surface hydrophobicity than 7S β-conglycinin [51]. In contrast, the higher abundance of 11S legumin in chickpeas compared to that in peas (Figure 1), did not contribute to higher surface hydrophobicity. This observation implied that variation in 11S amino acid sequence across species has a bigger impact on surface hydrophobicity than 7S/11S ratio.

Unlike surface hydrophobicity, variation in surface charge was limited across all pea and chickpea protein samples, with only a few minor statistical differences (Table 2). The extraction scale had no impact on the surface charge. However, ChPI and SU ChPI had a slightly but significantly higher net negative charge than PPI and SU PPI. While the surface charge of PPI was similar to previous reports [9,13,15], that of ChPI was higher than what was reported for both AE-IEP and salt-extracted ChPI [9]. This observation could be attributed to different extraction conditions (e.g., salt concentration, salt type), as well as the residual salt content in ChPI.

The surface hydrophobicity and charge of cSPI were also determined to better evaluate differences in functionality compared to pea and chickpea protein isolates. While cSPI had a relatively high surface hydrophobicity, it had a considerably higher net charge compared to all the samples. This balance between surface charge and surface hydrophobicity could explain the superiority of soy protein in certain functional properties as will be discussed.

### 3.6. Protein Solubility of PPI and ChPI as Impacted by Extraction Scale and in Comparison to Commercial Protein Ingredients 

Protein solubility of benchtop and scaled-up PPI and ChPI in comparison to commercial samples was assessed before and after heat treatment at both neutral and acidic pH (Table 3). At pH 7 SU PPI had significantly lower protein solubility than benchtop PPI, most likely due to partial protein denaturation and aggregation induced by thermal treatments during scale-up production, as discussed. Heating (80 °C for 30 min) at pH 7 resulted in a significant decrease in protein solubility of PPI but had no impact on the solubility of SU PPI. Since benchtop PPI was significantly less denatured and had significantly lower surface hydrophobicity (Table 2) than SU PPI, the heat treatment could have caused denaturation and polymerization that resulted in a significant reduction in solubility. 

In contrast, SU PPI was already partially denatured and had HMW polymers, potentially explaining the lack of impact of heat treatment on protein solubility. In comparison, ChPI and SU ChPI exhibited the highest protein solubility at pH 7.0 among all samples, regardless of heat treatment. The relatively lower surface hydrophobicity to charge ratio of ChPI and SU ChPI compared to PPI and SU PPI could have contributed to the observed differences in solubility. Given that chickpea protein compared to pea protein has a relatively higher proportion of legumin, which has a higher denaturation temperature (>80 °C) than vicilin, heating of ChPI and SU ChPI did not have a significant impact on protein solubility at pH 7. 

Although cSPI was completely denatured (Table 2), had a high degree of polymerization (Figure 1), and had a high surface hydrophobicity, it had an acceptable protein solubility at pH 7, which was attributed to its relatively high surface charge. In contrast, cPPI had the lowest solubility among all samples, due to being completely denatured, extensively polymerized, and having a high surface hydrophobicity to charge ratio compared to the other protein isolates. Given its mostly preserved protein structure, cPPC, on the other hand, had similar solubility to PPI and SU PPI. cChPC, however, had significantly lower solubility at pH 7 than ChPI and SU ChPI, due to protein denaturation and polymerization, as discussed. 

At pH 3.4, the net charge of the protein would be lower than that at pH 7 since the protein is closer to its average isoelectric point (pH 4.5). Nevertheless, both PPI and SU PPI exhibited good solubility at pH 3.4 (Table 3), which is significantly higher than all commercial samples including cSPI. This observation confirmed that the scaled-up production of PPI following SE-UF was successful in preserving the protein structure resulting in superior solubility to cPPI produced following AE-IEP, similar to the findings of Hansen et al. [7]. In contrast, ChPI had considerably low solubility at pH 3.4, similar to that of cSPI, while SU ChPI had good solubility similar to that of PPI and SU PPI. This observation can be explained by the charge load on the protein. Since the net charge of the protein is relatively low when the pH is close to the pI, a slightly elevated salt content could have a “salting out” effect, thereby decreasing the solubility of the protein. The ash content of the benchtop ChPI was significantly higher (*p* < 0.05) than that of SU ChPI (4.79% vs. 2.14% ash), thus potentially contributing to the observed difference in solubility. Similarly, Carbonaro et al. [52] reported a significant impact of salt content on the solubility of chickpea proteins at pH 4. cChPC, on the other hand, had inferior solubility compared to ChPI and SU ChPI at pH 3.4. This observation is again attributed to a higher degree of protein denaturation, higher surface hydrophobicity, and lower surface charge of cChPC compared to the produced isolates, in addition to the significantly higher ash content (Table 1). For the first time, this data confirmed that scaled-up production of ChPI following SE-UF can preserve protein structure and result in excellent solubility at both neutral and acidic pH, better than a commercial pea, chickpea, and soy protein ingredients. 

### 3.7. Gelation of PPI and ChPI as Impacted by Extraction Scale and in Comparison to Commercial Protein Ingredients

Thermally-induced gels of all protein samples exhibited excellent WHC at 15% protein concentration (>98%) (Data not shown). Regardless of gel strength at 15 % protein, each of the protein samples formed a relatively stable protein network that had minimum syneresis, with no apparent impact of protein source, extraction conditions, or extraction scale. On the other hand, significant differences in gel strength were observed among the samples at both 15% and 20% protein concentration (Figure 3). Gel strength was determined at both 15 and 20% protein concentration since commercial pea protein isolate typically either forms a weak gel or does not form a gel at all at 15% [7]. 

Among all protein ingredients, cSPI, at 15% protein concentration, had by far the highest gel strength, attributed to its good balance of protein-protein and protein-water interactions, as implied by its surface hydrophobicity to charge ratio. Another contributing factor is the higher 11S to 7S ratio and higher content of sulfhydryl groups in soy compared to pea and chickpea proteins [53]. In comparing chickpea to pea protein, ChPI and SU ChPI, specifically at 20% protein concentrations, had significantly higher gel strength than PPI and SU PPI, respectively. This observation again can be partially attributed to the higher 11S to 7S ratio in chickpeas compared to peas as evidenced by SDS-PAGE (Figure 1) and as previously reported [54]. In addition, the different 11S (legumin) variants (Figure 1, lanes 7 & 8) present in chickpeas could potentially have contributed to better gelation properties. Further research is needed to differentiate the composition of 11S variants in chickpeas compared to peas. 

Both SU PPI and SU ChPI, on the other hand, had significantly higher gel strength than their benchtop counterparts (Figure 3). The relatively higher surface hydrophobicity and partial denaturation (Table 2), as well as the presence of high molecular weight polymers (Figure 1) in scaled-up isolates potentially contributed to enhanced gel strength. Hydrophobic attractive forces will aid in bringing the protein molecules in closer proximity facilitating intermolecular disulfide linkages, thus strengthening the protein network. Compared to all the isolates, with the exception of cSPI, SU ChPI had the highest gel strength at both 15% and 20% protein. This observation confirmed that SE-UF can be scaled up to produce a chickpea protein isolate with better gelation potential than commercial pea protein isolate.

Among the commercial pea and chickpea protein samples, cPPC had significantly the highest gel strength at both 15% and 20% protein concentration. This observation is attributed mostly to the presence of starch, which acts as a good gelling agent [27,53]. During gel formation, heating the cPPC solution at 95 °C, above the gelatinization temperature of pea starch (64.2 °C) [55], contributed to enhanced gel strength. In contrast, cChPC did not outperform the protein isolates despite the presence of starch. The starch in cChPC was most likely pregelatinized and potentially retrograded [55]. The presence of pregelatinized starch, coupled with the denaturation state (Table 2), protein polymerization (Figure 1), and low solubility (Figure 3), had a compounded negative effect on the gel strength of cChPC. The impact of the processing steps employed during the production of cChPC needs to be investigated to identify the impact not only on the protein structure but also on the molecular characteristics of the residual chickpea starch [27] and fiber [28] in such a sample.

### 3.8. Emulsification Properties of PPI and ChPI as Impacted by Extraction Scale and in Comparison to Commercial Protein Ingredients

Minor statistical differences in emulsification properties were observed among the samples (Figure 4a–c). As expected, the EC of cSPI was superior among the protein isolates, with the exception of ChPI. This observation is attributed to a good balance between surface hydrophobicity and surface charge (Table 2). The EC of cPPC also was relatively high, which was attributed in part to the starch component [27]. On the other hand, the EC cChPC was comparable to that of cPPI.

While scaling up SE-UF production of PPI had no significant impact on EC, it did result in a significant decrease in EAI and ES. While partial denaturation aided in enhancing molecular flexibility and orientation at the interface, it could have contributed to attractive forces among the protein molecules on the interface, resulting in a slight reduction in emulsion stability. In contrast, all the measured emulsification properties of ChPI were adversely impacted by scaling up the extraction. The EC as well as ES of benchtop ChPI was the highest among the samples, owing to its well-preserved protein structure that had the highest enthalpy of denaturation, lowest surface hydrophobicity, and a relatively high surface charge compared to all pea and chickpea samples (Table 2). The shear induced by the homogenization employed during emulsion formation resulted in a partial unfolding of the native chickpea proteins in ChPI, allowing them to quickly migrate to the interface without precipitation, in contrast to already denatured and polymerized proteins. Withana-Gamage et al. [6] showed that benchtop ChPI had better emulsification properties (ES and EAI) than those of PPI, yet inferior to those of SPI, partially owing to the harsher extraction process (AE-IEP) that was adopted to produce ChPI. However, for a better understanding of the emulsification behavior of these proteins, an investigation of the molecular differences in the 11S and 7S proteins among chickpeas, pea, and soy is needed. Nevertheless, scaling up SE-UF production resulted in isolates of comparable emulsification properties to commercially available pea and chickpea protein ingredients (cPPI, cChPC), with SU PPI showing significantly higher EC. 

## 4. Conclusions

This study demonstrated that the selected SE-UF extraction conditions (0.5 M NaCl, 3.5 kDa cut-off membrane) can be successfully scaled up to produce ChPI and PPI with high protein purity, good protein yield, relatively preserved protein structure, and superior functionality to commercial counterparts (cChPC, cPPI). Specifically, this is the first study to evaluate the feasibility of scaling up the production of ChPI that had comparable or even better functional properties than both cSPI and cPPI. Specifically, SU ChPI had superior solubility at both neutral and acidic pH compared to cSPI and cPPI, and significantly outperformed cPPI in terms of gel strength. Accordingly, ChPI produced following the tested SE-UF process can be successfully incorporated in beverage applications and in food products requiring good gelling and water-holding properties. Additionally, the good gelation properties of SU ChPI could be leveraged for meat analogue applications. A comparative evaluation of the performance of both SU PPI and SU ChPI in various applications would be a natural follow-up study. Nevertheless, this work confirmed that SE-UF is scalable and thus should be commercially considered as an alternative protein extraction process for the production of pulse proteins with improved functional performance. 

## Figures and Tables

**Figure 1 foods-12-01694-f001:**
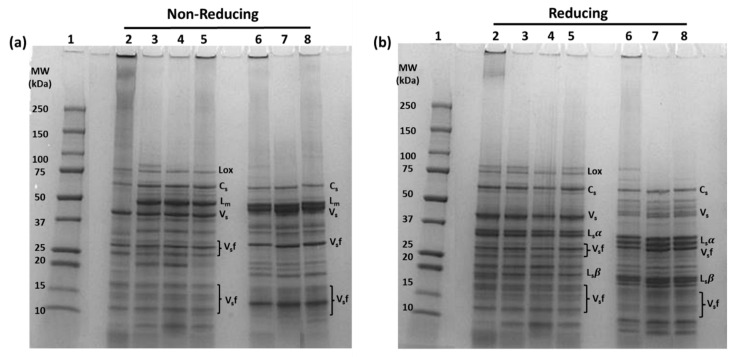
SDS-PAGE gel protein profile visualization of benchtop and scaled-up pea and chickpea protein isolates (PPI, SU PPI, ChPI, and SU ChPI), as well as commercial PPI, PPC, and ChPC under non-reducing (**a**) and reducing (**b**) conditions. Lane 1: Molecular weight (MW) marker; Lane 2, 3: cPPI and cPPC; Lane 4, 5: PPI and SU PPI, Lane 6: cChPC; Lane 7, 8: ChPI and SU ChPI. Lox: lipoxygenase; C_s_: convicilin subunits; L_m_: legumin monomer; V_s_: vicilin subunits; L_s_α: legumin acidic subunits, L_s_β: legumin basic subunits; V_s_f: vicilin subunit fractions due to post-translational cleavages.

**Figure 2 foods-12-01694-f002:**
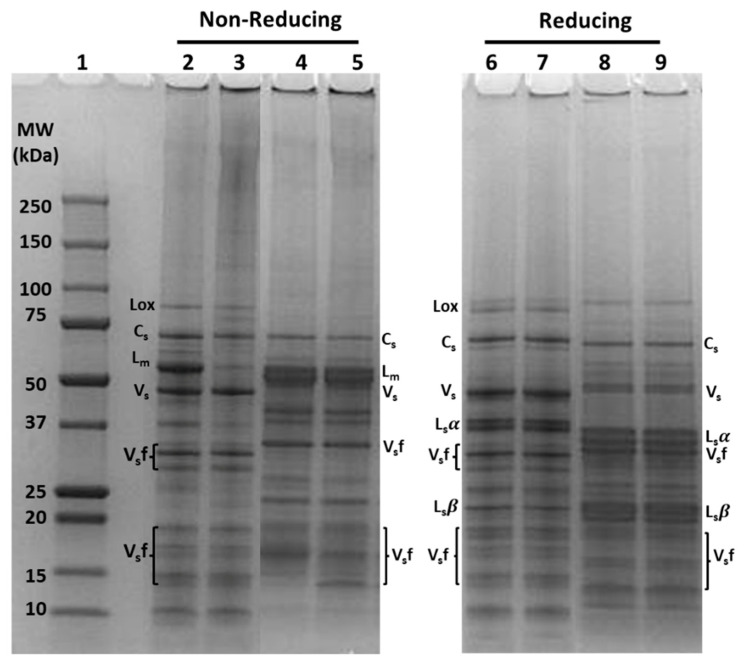
SDS-PAGE gel protein profile visualization of high solid (HS) and low solid (LS) fractions of SU PPI compared to that of cPPI and SU ChPI compared to that of cChPC under non-reducing (lane 2–5) and reducing (lane 6–9) conditions. Lane 1: Molecular weight (MW) marker; Lane 2, 6: SU PPI HS; Lane 3, 7: SU PPI LS; Lane 4, 8: SU ChPI HS; Lane 5, 9: SU ChPI LS. Lox: lipoxygenase; C_s_: convicilin subunits; L_m_: legumin monomer; V_s_: vicilin subunits; L_s_α: legumin acidic subunits, L_s_β: legumin basic subunits; V_s_f: vicilin subunit fractions due to post-translational cleavages.

**Figure 3 foods-12-01694-f003:**
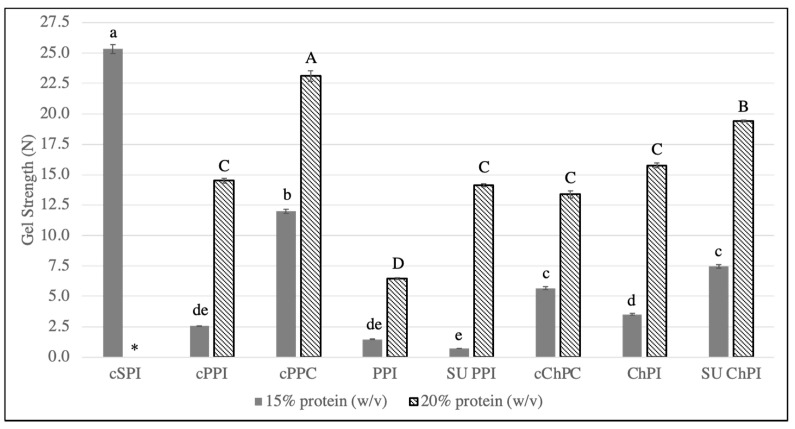
Protein gel strength of benchtop and scaled-up pea and chickpea protein isolates (PPI, SU PPI, ChPI, SU ChPI), as well as commercial SPI, PPI, PPC, and cChPC. An asterisk (*) denotes sample not analyzed due to high viscosity during sample solubilization. Error bars represent standard error (n = 3). Lowercase and uppercase letters above the bars denote significant differences among the samples evaluated at 15% and 20% protein concentration, respectively, according to the Tukey-Kramer multiple means comparison test (*p* < 0.05).

**Figure 4 foods-12-01694-f004:**
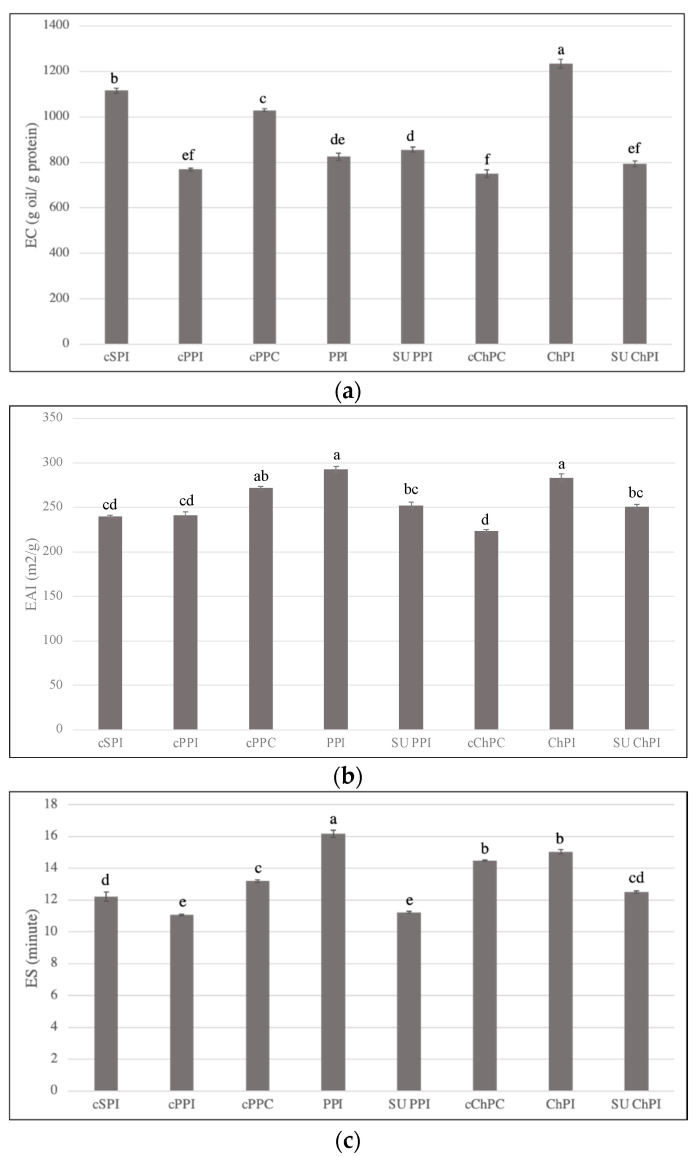
Protein emulsion capacity, EC, (**a**) emulsion activity index, EAI, (**b**) and emulsion stability, ES, (**c**) of benchtop and scaled-up pea and chickpea protein isolates (PPI, SU PPI, ChPI, SU ChPI), as well as commercial SPI, PPI, PPC, and cChPC. Error bars represent standard error (n = 3). Lowercase letters above the bars denote significant differences among the samples, according to the Tukey-Kramer multiple means comparison test (*p* < 0.05).

**Table 1 foods-12-01694-t001:** Protein, ash, and color (L * a * b * and ∆E) of benchtop and scaled-up pea and chickpea protein isolates (PPI, SU PPI, ChPI, and SU ChPI), as well as commercial SPI, PPI, PPC and ChPC.

Samples	Protein (%)	Ash (%)	Color			
			L *	a *	b *	∆E ^1^
cSPI	90.1 ± 0.07 ^d2^	4.16 ± 0.14 ^d^	86.7 ± 0.05 ^e^	−0.23 ± 0.03 ^e^	14.3 ± 0.08 ^d^	
cPPI	79.5 ± 0.20 ^e^	5.61 ± 0.03 ^a^	86.7 ± 0.04 ^e^	0.51 ± 0.02 ^c^	17.3 ± 0.11 ^b^	
cPPC	52.4 ± 0.18 ^h^	5.14 ± 0.10 ^bc^	92.2 ± 0.25 ^b^	−1.21 ± 0.04 ^g^	13.7 ± 0.17 ^de^	
PPI	92.9 ± 0.17 ^b^	2.14 ± 0.08 ^e^	84.7 ± 0.03 ^f^	1.07 ± 0.01 ^b^	19.4 ± 0.04 ^a^	3.00 ± 0.11 ^B3^
SU PPI	90.9 ± 0.03 ^d^	2.39 ± 0.09 ^e^	88.4 ± 0.01 ^d^	0.01 ± 0.03 ^d^	12.5 ± 0.03 ^f^	5.04 ± 0.07 ^A^
cChPC	56.4 ± 0.23 ^f^	5.27 ± 0.11 ^ab^	92.9 ± 0.08 ^a^	−0.62 ± 0.03 ^f^	9.05 ± 0.10 ^g^	
ChPI	91.9 ± 0.15 ^c^	4.79 ± 0.07 ^c^	85.2 ± 0.04 ^f^	2.00 ± 0.05 ^a^	15.2 ± 0.43 ^c^	10.20 ± 0.28 ^α4^
SU ChPI	94.0 ± 0.23 ^a^	2.14 ± 0.06 ^e^	90.9 ± 0.00 ^c^	−1.02 ± 0.09 ^g^	13.5 ± 0.06 ^de^	4.91 ± 0.01 ^β^

^1^ Total color difference (ΔE) between each produced isolate and its respective commercial reference (cPPI and cChPC); ^2^ Lowercase letters denote significant differences among the means (n = 3) in each column, according to the Tukey–Kramer multiple means comparison test (*p* < 0.05); ^3^ Uppercase letters indicate significant differences between the ∆E of PPI and SU PPI in comparison to cPPI; ^4^ Greek alphabets indicate significant differences between the ∆E of ChPI and SU ChPI in comparison to cChPC, according to the student’s unpaired *t*-test (*p* < 0.05).

**Table 2 foods-12-01694-t002:** Denaturation temperature and enthalpy, surface hydrophobicity, and surface charge of benchtop and scaled-up pea and chickpea protein isolates (PPI, SU PPI, ChPI, and SU ChPI), as well as commercial SPI, PPI, PPC, and ChPC.

Samples	Denaturation Temperature and Enthalpy			Surface Properties	
	Denaturation Temperature (Td)		Total Enthalpy of Denaturation (ΔH)	Surface Hydrophobicity	Surface Charge
	°C		J g^−1^	RFI	mV
	β-conglycinin	Glycinin			
cSPI	*^1^	*	*	10,820.3 ± 530.3 ^b^	−41.3 ± 0.20 ^a^
	Vicilin (7S)	Legumin (11S)			
cPPI	*	*	*	13,821.7 ± 434.4 ^a^	−30.2 ± 0.13 ^bc^
cPPC	85.5 ± 0.02 ^a2^	94.4 ± 0.12 ^b^	2.30 ± 0.04 ^f^	7895.7 ± 271.8 ^cd^	−27.9 ± 0.30 ^cd^
PPI	82.6 ± 0.13 ^b^	88.1 ± 0.03 ^e^	10.9 ± 0.50 ^b^	6564.4 ± 129.5 ^d^	−26.2 ± 0.34 ^d^
SU PPI	82.6 ± 0.13 ^b^	89.9 ± 0.16 ^d^	5.45 ± 0.07 ^d^	14,199.7 ± 105.9 ^a^	−27.2 ± 0.07 ^d^
cChPC	81.5 ± 0.09 ^c^	99.6 ± 0.02 ^a^	3.77 ± 0.09 ^e^	13,317.0 ± 450.4 ^a^	−25.7 ± 0.33 ^d^
ChPI	81.6 ± 0.08 ^c^	89.9 ± 0.11 ^d^	16.8 ± 0.54 ^a^	4491.1 ± 157.9 ^e^	−30.8 ± 0.15 ^b^
SU ChPI	80.5 ± 0.07 ^d^	90.8 ± 0.17 ^c^	8.61 ± 0.14 ^c^	8973.3 ± 186.5 ^c^	−30.9 ± 0.21 ^b^

^1^ An asterisk (*) indicates the absence of endothermic peaks due to complete protein denaturation; ^2^ Lowercase letters indicate significant differences among the means (n = 3) in each column, according to the Tukey–Kramer multiple means comparison test (*p* < 0.05).

**Table 3 foods-12-01694-t003:** Protein solubility of benchtop and scaled-up pea and chickpea protein isolates (PPI, SU PPI, ChPI, and SU ChPI), as well as commercial SPI, PPI, PPC, and ChPC.

Samples	Percent Protein Solubility (5% Protein)			
	pH 7.0		pH 3.4	
	Non-Heated	Heated ^1^	Non-Heated	Heated
cSPI	66.8 ± 0.40 ^d2^	78.5 ± 0.39 ^b^*^3^	24.9 ± 0.53 ^c^	39.1 ± 0.11 ^b^*
cPPI	29.5 ± 0.85 ^e^	57.1 ± 0.64 ^e^*	11.6 ± 0.49 ^d^	17.5 ± 0.79 ^c^*
cPPC	84.3 ± 0.17 ^b^	67.4 ± 0.30 ^cd^*	23.8 ± 0.92 ^c^	20.2 ± 1.18 ^c^
PPI	84.8 ± 0.22 ^b^	57.4 ± 0.87 ^e^*	82.7 ± 0.54 ^a^	85.2 ± 0.12 ^a^*
SU PPI	68.9 ± 0.42 ^cd^	69.9 ± 0.90 ^c^	71.7 ± 0.25 ^b^	80.3 ± 0.48 ^a^*
cChPC	70.7 ± 0.31 ^c^	66.7 ± 0.47 ^d^*	12.7 ± 0.20 ^d^	16.2 ± 0.22 ^c^*
ChPI	96.2 ± 0.09 ^a^	94.1 ± 0.52 ^a^	26.1 ± 0.58 ^c^	34.5 ± 0.70 ^b^*
SU ChPI	94.3 ± 0.69 ^a^	92.5 ± 0.76 ^a^	71.5 ± 3.00 ^b^	82.8 ± 3.00 ^a^*

^1^ Heated at 80 °C for 30 min; ^2^ Lowercase letters denote significant differences among the means (n = 3) in each column, according to the Tukey-Kramer multiple means comparison test (*p* < 0.05); ^3^ An asterisk denotes significant differences between non-heated and heated samples (*p* < 0.05), according to the student’s unpaired *t*-test (*p* < 0.05).

## Data Availability

Data is available upon reasonable request.

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
