# Peer review of "Salt Solubilization Coupled with Membrane Filtration-Impact on the Structure/Function of Chickpea Compared to Pea Protein"

_foods, 2023, doi:10.3390/foods12081694_

Round 1

Reviewer 1 Report

The manuscript is well written though minor suggestions needs. 

In Abstract Keywords: Keywords needs to be a single and crispy word which needs to attract the readers. For eg:- "protein structure and functionality" is normal can be changed. 

 Introduction can be elaborated (may be 1 more para) about the significance of salt extraction coupled with ultrafiltration. 

Also, Aim of the research can be improved by adding the novelty of the research. 

In Methods "2.5 Color measurement" can you elaborate "commercial reference was also calculated". 

In Results: If possible try to change or include any figures replacing any table for wider attraction. 

Conclusion can be included with the future prespective of the research. 

Check the italics of binomial names, Some cited references are repeated can be changed with recent reference. 

Reviewer 2 Report

The study with the title Salt solubilization coupled to membrane filtration - Impact on chickpea structure/function compared to pea protein, shows that salt extraction coupled with ultrafiltration (0.5 M NaCl, membrane cut-off 3.5 620 kDa) can be successfully scaled up to produce chickpea protein isolates (ChPI) and pea protein isolates (PPI) with high protein purity, good protein yield, relatively conserved protein structure and superior functionality compared to commercial counterparts (cChPC and cPPI). Therefore, from a commercial point of view, this method should be considered as an alternative protein extraction process for the production of pulse proteins with improved functional performance.

Reviewer 3 Report

The manuscript “Salt Solubilization Coupled with Membrane Filtration-Impact on the Structure/Function of Chickpea compared to Pea Protein” by Brigitta P. Yaputri, Fan Bu and Baraem P. Ismail documents the preparation of chickpea protein isolate by salt solubilization coupled with ultrafiltration and comparing the produced product with pea protein isolate.

My notes and observations:

* In section 2.2. it says that different conditions were tried, but no results are given, no numerical values are given, so the chosen conditions are not strongly justified.

* In all tables, all values should be presented with errors when evaluating significant differences.

* In the authors' previous work, based on which SE-UF is performed, and which is often cited in this paper ([7]-line 663-664), compared to this paper, gives somewhat different values of the same parameters, e.g. protein solubility of cSPI, cPPI, PPI, SU PPI, emulsion stability values, and the inconsistency of other parameters make us question the reliability of the results. The flour from which PPI and SU PPI are obtained is the same in both the previous work and this one. True, cPPI is another manufacturer. Since the parameters of commercial products are different, depending on the manufacturer and various conditions, the question is how appropriate is the comparison with them?

* Does it make sense to use cCHPC for comparison? Is there really no commercial cCHPI available?

* Line 594 – “were observed among the samples (Table 5)…” - there are only 3 tables in the work.

* Line 678 - Capitalize only the first letter of the last name.

* More than half of the cited articles are older than 5 years. However, recent literature is also cited sufficiently. DOI numbers are provided for all publications in the bibliography except for one (29, lines 729-732). Please do the same.
